# Conserved and unique features of terminal telomeric sequences in ALT-positive cancer cells

**Benura Azeroglu, Wei Wu[†], Raphael Pavani, Ranjodh Singh Sandhu, Tadahiko Matsumoto[‡], André Nussenzweig, Eros Lazzerini-Denchi***

Laboratory of Genome Integrity, National Cancer Institute, National Institutes of Health, Bethesda, United States

**\*For correspondence:**
eros.lazzerinidenchi@nih.gov

**Present address:** [†]Key Laboratory of Multi-Cell Systems, Shanghai Institute of Biochemistry and Cell Biology, Center for Excellence in Molecular Cell Science, Chinese Academy of Sciences, Shanghai, China; [‡]Kyoto University, Kyoto, Japan

**Competing interest:** The authors declare that no competing interests exist.

## eLife Assessment

This study demonstrates the application of END-seq, originally developed to study genomewide DNA double-strand breaks, to telomere biology; the work packs a punch, concisely demonstrating the utility of this approach and the new insights that can be gained. The authors confirm that telomeres in telomerase-positive cells terminate with 5'-ATC in a Pot1-dependent manner and demonstrate that this principle holds true in telomerase-negative ALT cells as well. S1-END-seq is similarly developed for telomeres, showing that ALT cells harbor several regions of ssDNA. The study is well-executed and **convincing**, the new insights are **fundamental** and **compelling**, and the optimized END-seq approaches will be widely utilized. The work will prompt additional studies that the reviewers look forward to, including combining telomeric END-seq with long-read sequencing to address the distribution and origin of variant telomere repeats and ssDNA along telomeres in ALT and telomerase-positive settings.

## Abstract

A significant portion of human cancers utilize a recombination-based pathway, alternative lengthening of telomeres (ALT), to maintain telomere length. Targeting the ALT is of growing interest as a cancer therapy, yet a substantial knowledge gap remains regarding the basic features of telomeres in ALT-positive cells. To address this, we adopted END-seq, an unbiased sequencing-based approach, to define the identity and regulation of the terminal sequences of chromosomes in ALT cells. Our data reveal that the terminal portions of chromosomes in ALT cells contain canonical telomeric sequences with the same terminus bias (-ATC) observed in non-ALT cells. Furthermore, as reported for non-ALT cells, POT1 is required to preserve the precise regulation of the 5′ end in cells that maintain telomere length using the ALT pathway. Thus, the regulation of the terminal 5′ of chromosomes occurs independently of the mechanism of telomere elongation. Additionally, we employed an S1 endonuclease-based sequencing method to determine the presence and origin of single-stranded regions within ALT telomeres. These data shed light on conserved and unique features of ALT telomeres.

## Introduction

Telomeres are the natural ends of eukaryotic chromosomes and are critical for maintaining chromosome integrity (*Palm and de Lange, 2008*). Human telomeres consist of 5–15 kilobases of repetitive DNA sequences (5'-TTAGGG-3'), which serve as a binding platform for an essential protective protein complex named shelterin (*Palm and de Lange, 2008*). The terminal portion of chromosomes is composed of a single-stranded 3′ overhang of approximately 200 nucleotides, generated by the

interplay between DNA replication and end-processing, and regulated by the telomere-associated protein POT1 (*Cai and de Lange, 2023*; *Lim and Cech, 2021*). It is noteworthy that the generation of the 3′ overhang differs between leading and lagging strand telomeres. At leading strand telomeres, the newly synthesized DNA terminates in a blunt end and requires post-replicative resection by nucleases such as Apollo and Exo1 and a fill-in action promoted by CST complex to create the 5′ overhang (*Wu et al., 2012*). In contrast, lagging strand synthesis inherently generates a 3′ overhang due to Okazaki fragment processing, where removal of the terminal RNA primer leaves a single-stranded gap that is partially filled in by DNA polymerase α-primase with the aid of the CST complex (*Takai et al., 2024*; for reviews, see *Cai and de Lange, 2023*; *Lim and Cech, 2021*). Progressive loss of telomeric repeats during DNA replication leads to telomere shortening and represents a barrier to unlimited proliferation (*Feldser and Greider, 2007*; *Greenberg et al., 1998*; *Greider, 1996*; *Perera et al., 2008*). To overcome this tumor suppressive barrier, ~85% of cancers activate telomerase, a reverse transcriptase, to synthesize new telomeric repeats (*Shay and Bacchetti, 1997*; *Kim et al., 1994*). The remaining ~15% of cancers engage the alternative lengthening of telomeres (ALT) pathway, a recombination-based mechanism to elongate telomeres using existing telomeric DNA as a template (*Dunham et al., 2000*; *Bryan et al., 1997*). Major advancements in our understanding of the ALT pathway at the molecular level have been made over recent years, shedding light on the molecular pathways involved in telomere elongation as well as the genetic vulnerabilities of ALT-positive cells (*Jiang et al., 2024*; *Lu et al., 2019*; *Pan et al., 2019*; *Panier et al., 2019*; *Silva et al., 2019*; *Loe et al., 2020*; *Dilley et al., 2016*; *Barroso-González et al., 2020*; *Kaminski et al., 2022*; *Lynskey et al., 2024*; *Azeroglu et al., 2025*; for reviews, see *Bhargava et al., 2022*; *Claude and Decottignies, 2020*; *Loe et al., 2023*; *O'Sullivan and Greenberg, 2025*).

Due to the repetitive nature of telomeres, some fundamental questions about the composition of ALT telomeres remain unanswered. Specifically, it is unclear if 5′ telomere ends in ALT cells differ from telomerase-positive cells, whether they follow the same rules for end protection, and whether ALT telomeres contain any unique secondary structures. It is foreseeable that telomeres in telomerase-positive cells and ALT cells differ, as ALT can generate complex and non-canonical telomeric sequences, and secondary structures through recombination-based mechanisms. Interestingly, positive staining for the single-strand binding protein RPA and native FISH staining (ssTelo) of telomeres suggested the presence of single-stranded DNA regions in ALT-positive cells (*Loe et al., 2020*; *Grudic et al., 2007*). However, the source or extent of ssDNA in ALT telomeres remains unclear. Intriguingly, EM data and rolling-circle amplification-based assays suggest that ALT frequently contain circular extrachromosomal telomeric DNA termed C-circles (*Henson et al., 2009*; *Cesare and Griffith, 2004*).

Here, we report a novel approach to studying ALT telomeres using the high-resolution and unbiased method known as END-seq (*Canela et al., 2016*). Using this method, we find that the terminal sequence of ALT chromosomes consists of canonical telomeric repeats, shares the same bias in the 5′ terminus as telomeres maintained by telomerase, and terminates with ATC-5′. Furthermore, we demonstrated that this bias is influenced by the shelterin component, POT1.

Secondly, to assess the presence of ssDNA at ALT telomeres, we used S1 nuclease, an enzyme that specifically degrades single-stranded DNA including the single-stranded regions in duplex DNA. Using S1 nuclease treatment coupled with END-seq (S1-END-seq) (*Matos-Rodrigues et al., 2022*), we show that ALT telomeres have tracts of ssDNA and the presence of ssDNA in telomeres can be used to distinguish ALT-positive from ALT-negative cells.

## Results

### END-seq efficiently captures 5′ chromosome termini

To study the terminal portion of chromosome, we employed an unbiased and high-resolution approach developed to map and investigate double-strand breaks (DSBs): END-seq (*Canela et al., 2016*). In this method, biotin-labeled adaptors are ligated to DSBs, which enable their purification and subsequent sequencing by next-generation sequencing (NGS). Based on the directionality of DNA sequencing (5′ to 3′), DSBs are identified as unidirectional reads that originate from the lesion site on either side of the break (*Figure 1A*). Chromosome ends resemble one-ended DSBs, which by END-seq should be detected as unidirectional reads corresponding to the 5′ telomeric strand, the C-rich strand. To test this hypothesis, we performed END-seq on two telomerase-positive human cell lines, HeLa and

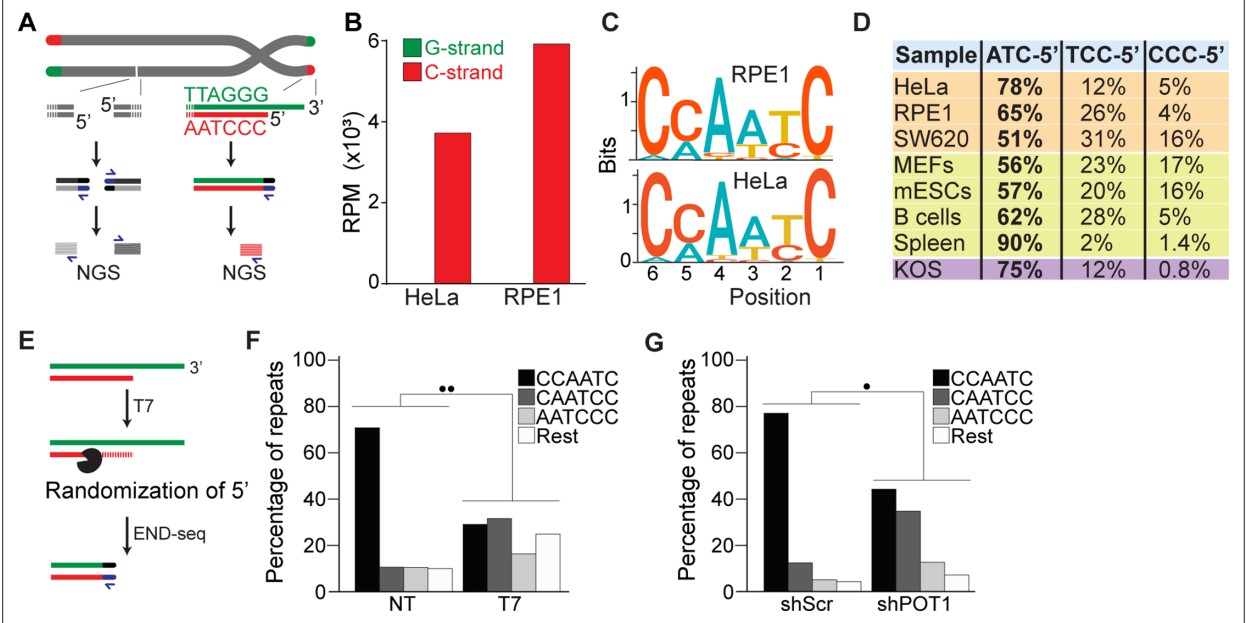

**Figure 1.** END-seq successfully captures the 5′ termini of human and mouse chromosomes. (**A**) Schematic representation of the END-seq method applied to double-strand breaks (DSBs) (left) and natural chromosome ends (right). Blunted dsDNA ends are ligated to biotinylated adaptors, purified, and subsequently sequenced by next-generation sequencing (NGS). Reads originating from DSBs align to either side of the break (left), while telomeric reads align only to the 5′ C-rich template (red). (**B**) Number of reads containing at least four consecutive telomeric repeats corresponding to the G-rich (green) and C-rich (red) strands in HeLa and RPE1 cell lines. Telomeric reads are normalized to the total number of reads identified by END-seq and are represented as reads per million (RPM). (**C**) Sequence logo representing the conservation (bits) of the last 6 nucleotides at the 5′ end of chromosomes in HeLa and RPE1 cells. (**D**) Distribution of the three most frequent telomeric repeats across human cell lines (orange), mouse cells (yellow), and a canine cell line (purple). (**E**) Schematic representation of T7-mediated randomization of the 5′ termini. (**F**) Percentage of telomeric reads that have the indicated sequence as a 5′ end. The following sequences (CCCAAT-5′, TCCCAA-5′, and ATCCCA-5′) are grouped and labeled as 'Rest'. Kullback–Leibler divergence (KL divergence) analysis was used to compare the distributions of the individual conditions. KL divergence between 0.25 and 0.375 is represented by two dots. Prior to the END-seq protocol, cells were either left untreated (NT) or were treated with T7. (**G**) Percentage of telomeric reads is displayed as described in (**F**). Cells expressing either a nontargeting shRNA (shScr) or an shRNA targeting POT1 (shPOT1) were harvested 3 days post induction and analyzed by END-seq. KL divergence between 0.125 and 0.25 is represented by one dot (●).

The online version of this article includes the following figure supplement(s) for figure 1:

**Figure supplement 1.** Analysis of telomeric and centromeric reads and 5′ termini randomization.

telomerase-expressing RPE1. Sequencing reads containing four consecutive TTAGGG repeats (G-rich) or CCCTAA repeats (C-rich) were defined as "telomeric reads" and were normalized to the total number of reads obtained in each individual library. Strikingly, virtually all (99.9%) telomeric END-seq reads correspond to the C-rich strand, in agreement with the fact that chromosome ends are one-ended DSBs (*Figure 1B*). To exclude possible artifacts derived from the repetitive nature of telomeric reads, we performed a similar analysis of reads corresponding to pan-centromeric repeats; this analysis shows that although the fraction of their presence in the human genome was similar (*Figure 1—figure supplement 1A*), the pan-centromeric sequences are not enriched to the same level as telomeres nor show any strand bias when detected (*Figure 1—figure supplement 1B*).

Next, we performed a motif analysis on the C-rich telomeric END-seq reads to determine the prevalent terminal chromosomal sequence. This analysis shows that the vast majority (78% and 65% for HeLa and RPE1, respectively) of chromosomes end with ATC-5′ (*Figure 1C and D*). Strikingly, this is in agreement with previous data obtained using a PCR-based approach termed STELA on defined chromosome ends in HeLa cells (*Sfeir et al., 2005*). A major advantage of END-seq over STELA is that it can be applied to telomeres of any length and chromosome location without the need of specific sub-telomeric primers. This excludes possible biases and allows the analysis of telomeres from any cell type and origin. As proof of principle, we expanded the analysis to a collection of telomerase-positive human cell lines, as well as mouse and dog cell lines, and mouse tissues. In these experiments, we found that chromosome ends show a strong bias for ATC-5′ as a terminal sequencing (*Figure 1D*), in

line with previous findings (*Tesmer et al., 2023*; *Smoom et al., 2023*), suggesting a conserved mechanism for its generation.

To further validate END-seq as a method to identify terminal telomeric repeats, we tested whether 5′–3′ T7 exonuclease treatment prior to END-seq would be sufficient to change the bias of the ATC-5′ sequence (*Figure 1E*). Since T7 exonuclease preferentially degrades 5′ ends of double-stranded DNA while leaving 3′ single-stranded overhangs intact, we expected that its treatment would randomize the 5′ telomeric ends by progressively resecting them, thereby disrupting the original sequence bias. If END-seq was biased or not sufficiently sensitive, T7 treatment would not significantly alter the detected sequence distribution. As shown in *Figure 1F* and *Figure 1—figure supplement 1C*, T7-treated samples display an altered distribution of telomeric permutations. Quantitative measurement of the difference in distribution of possible telomeric permutations in the presence or absence of T7 treatment was performed using the Kullback–Leibler divergence (KL divergence) score. The result of this analysis indicated that the samples are extremely divergent (●● indicates a KL divergence value between 0.375 and 0.5). To further evaluate the distribution changes between these conditions, we compared the reads obtained from these samples with *in silico*-generated control reads representing various degrees of randomization from the ATC-5′ sequences (see *Figure 1—figure supplement 1D and E*). Using the KL divergence score, we determined that T7-randomized samples were most similar to a population where 100% of the reads were equally randomized. In comparison, in the untreated samples, approximately 70% of the reads correspond to the perfect 'canonical' CCAATC-5′ sequence. These data indicate END-seq as a method to sensitively detect terminal telomeric repeats while excluding the potential artifacts.

Further, we investigated whether we could detect in vivo alterations of the terminal repeat, such as those reported upon depletion of the shelterin component, POT1 (*Hockemeyer et al., 2005*). To deplete POT1, we generated a population of HeLa cells harboring a doxycycline-inducible POT1 short hairpin RNA (shPOT1-1) construct. Doxycycline treatment resulted in reduced POT1 expression, confirming efficient knockdown (*Figure 1—figure supplement 1G*). In agreement with previously published data, END-seq analysis showed that POT1-depleted cells have a significant reduction in terminal repeats ending with the canonical ATC-5′ sequence (*Figure 1F*). Comparison with control reads generated *in silico* reveals that approximately 45% of reads from POT1-depleted cells are derived from random permutations of the telomeric repeat (*Figure 1—figure supplement 1F*).

Lastly, we tested how depletion of either or both POT1 paralogs affected the 5′ termini of mouse telomeres. To this end, we engineered mouse embryonic stem cells (mESCs) expressing a FKBP-tagged POT1a protein (POT1a$^{dTAG/dTAG}$) that is degraded when cells are treated with dTAG13 compound (*Figure 2—figure supplement 1A–C*). Using this cell line, we then generated cell lines lacking POT1b through CRISPR knockout (POT1a$^{dTAG/dTAG}$ POT1b$^{-/-}$) (*Figure 2—figure supplement 1D–F*). This strategy allowed us to examine the 5′ termini of POT1-proficient ESCs, as well as systematically analyze the effect of either single or double POT1 depletion in a genetically defined background. END-seq analysis revealed that while depletion of POT1a altered 5′ termini, depletion of POT1b had no effect (*Figure 2A and B*). Using the KL divergence score, we determined that in the POT1-proficient and POT1b-depleted samples, approximately 70% and 85% of reads, respectively, corresponded to the perfect canonical ends. In contrast, in POT1a-depleted samples, this proportion was around 25%, regardless of POT1b status. This striking result highlights a fundamental difference between POT1a and POT1b, demonstrating that POT1a plays a dominant role in maintaining proper 5′ end processing, consistent with recently published work (*Tesmer et al., 2023*). The stark contrast between their effects underscores the distinct functions of these paralogs and reveals an exciting new layer of telomere regulation. Collectively, these data demonstrate the effectiveness of END-seq to study terminal repeats of chromosomes.

## The terminal 5′ sequence in ALT-positive cells is canonical

Having established that END-seq efficiently captures the terminal repeat of chromosome ends, we sought to investigate whether ALT-positive cells have the same tight regulation of terminal repeats as telomerase-positive cells. We performed END-seq on three ALT-positive cell line: G292, SAOS2, and U2OS. Our analysis shows that in ALT cells, the vast majority of telomeric reads (82–98% across 3 cell lines and 11 biological repeats) correspond to the C-rich strand (*Figure 3A–C*), consistent with non-ALT cells and the expected pattern for a one-ended-DSB. Motif analysis revealed that the majority

of telomeric reads in all three tested cell lines start with the canonical ATC-5′ sequence (*Figure 3D*, *Figure 3—figure supplement 1A–C*). These data suggest that regulation of the terminal 5′ of ends of chromosomes occurs independently of the mechanism of telomere elongation. To test this hypothesis, we depleted POT1 in U2OS cells using two different doxycycline-inducible shRNAs (*Figure 3—figure supplement 1D*). POT1 depletion resulted in formation of 53BP1 foci (*Figure 3E and F*), consistent with POT1's established role in suppressing the DNA damage response at chromosome ends (*Hockemeyer et al., 2005*; *Denchi and de Lange, 2007*). Next, we performed END-seq to evaluate the effect of POT1 depletion on the terminal sequence. The data show that, similarly to telomerase-positive HeLa cells, depletion of POT1 dramatically altered the 5′ natural termini in ALT-positive cell lines (*Figure 3G*).

Next, we asked whether ALT cells are more likely to contain variant telomeric repeats (VTRs) in the terminal portions of chromosome ends. VTRs have been shown to occur frequently in telomeric reads isolated from ALT-positive cells (*Conomos et al., 2012*). To establish whether VTRs in ALT cells are contained within terminal repeats, we assessed the frequency of non-canonical telomeric repeats within the first 30 nucleotides from the 5′ of the Illumina reads from U2OS (1.93%) and HeLa (1.93%) cells (*Figure 3—figure supplement 1E*). This result shows that in ALT-positive cells, VTRs do not occur more frequently than non-ALT cells within the very terminal portion of chromosome ends. Due to the short read lengths used in this approach, we cannot determine whether VTRs in ALT cells are specifically excluded from the terminal region of chromosome ends.

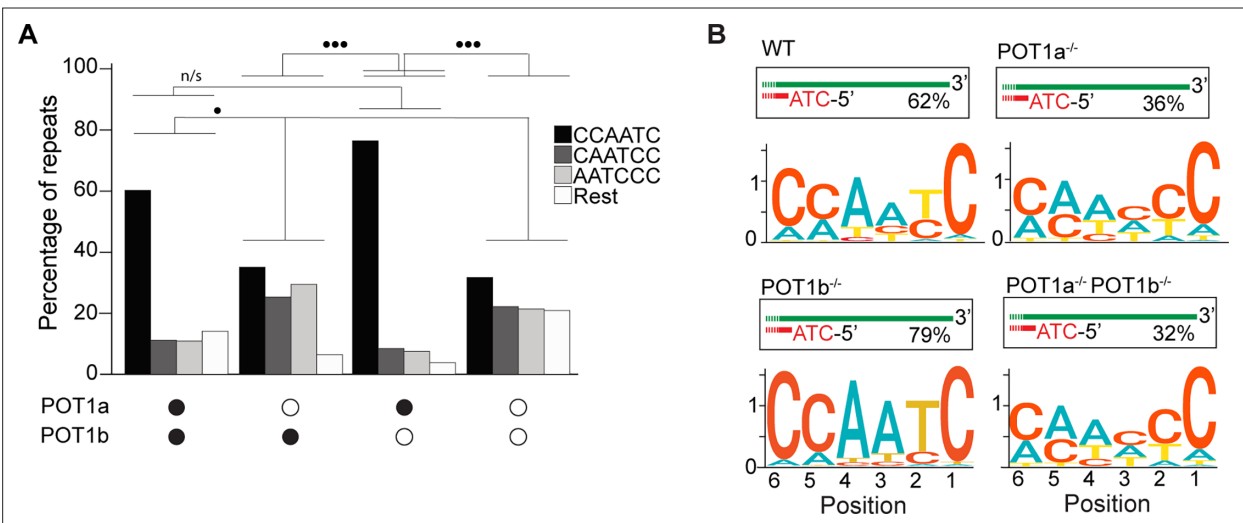

**Figure 2.** POT1a is the key regulator of 5′ telomere end processing in mice, distinct from POT1b. (**A**) Comparison of the effect of single and double depletion of POT1a and POT1b on the percentage of telomeric reads with the indicated 5′ end sequences. The following sequences (CCCAAT-5′, TCCCAA-5′, and ATCCCA-5′) are grouped and labeled as 'Rest'. Kullback–Leibler divergence (KL divergence) analysis was used to compare the distributions across conditions. KL divergence between 0.25 and 0.375 is represented by two dots. (**B**) Sequence logo illustrating the conservation (in bits) of the last 6 nucleotides at the 5′ end of chromosomes in POT1-proficient, POT1a-depleted, POT1b-depleted, and POT1a/POT1b double-depleted mouse embryonic stem cells (mESCs).

The online version of this article includes the following source data and figure supplement(s) for figure 2:

**Source data 1.** Original agarose gels and western blots corresponding to *Figure 3—figure supplement 1B, C, and E*, with relevant bands and treatments indicated.

**Source data 2.** Original files for agarose gels and western blot corresponding to *Figure 3—figure supplement 1B, C, and E*.

**Figure supplement 1.** CRISPR/Cas9-mediated endogenous knock-in and knockout of POT1a and POT1b in mouse embryonic stem cells (mESCs) with functional validation.

**Figure supplement 1—source data 1.** Original agarose gels and western blots corresponding to *Figure 2—figure supplement 1B, C and E*, with relevant bands and treatments indicated.

**Figure supplement 1—source data 2.** Original files for agarose gels and western blot corresponding to *Figure 2—figure supplement 1B, C and E*.

## Frequent occurrence of ssDNA in ALT telomeres

Having demonstrated that the 5′ terminal repeats in ALT cells are indistinguishable from non-ALT cells, we next asked whether the presence of single-stranded DNA (ssDNA) could be a unique feature of telomeric DNA in ALT-positive cells. Previous work provided indirect evidence for the presence of single-stranded telomeric DNA in ALT-positive cells, such as staining for RPA at telomeres, and FISH under native conditions (*Figure 4—figure supplement 1A–D*; *Loe et al., 2020*; *Grudic et al., 2007*). To test whether telomeres in ALT cells contain ssDNA, we employed S1-END-seq as a direct measure for the presence of ssDNA (*Matos-Rodrigues et al., 2022*). For this method, the S1 nuclease is used to convert any ssDNA into DSBs which are detected by END-seq. We anticipated that if telomeres do not contain any single-stranded telomeric DNA in addition to the 3′ overhang, telomeric S1-END-seq reads will be indistinguishable from the signal obtained using standard END-seq, corresponding to one-ended DNA ends (*Figure 4A*). However, S1-mediated cleavage of ssDNA within the internal telomeric tract will generate two-ended DSBs within telomeres. A DSB within telomeres would result in telomeric reads corresponding both to the C-rich and the G-rich strands (*Figure 4A*). To test this prediction, we performed S1-END-seq on a panel of ALT-positive and ALT-negative (*Figure 4B*, *Figure 4—figure supplement 1E*). S1-END-seq in ALT-negative cells revealed only C-strand telomeric reads excluding the presence of a significant amount of ssDNA within the telomeric tract in these cells (*Figure 4B*, *Figure 4—figure supplement 1E*). By contrast, performing S1-END-seq on a panel of ALT-positive cell lines revealed that G-rich and C-rich telomeric reads were detected at similar frequencies (*Figure 4B*, *Figure 4—figure supplement 1E*). These data show that telomeres in ALT-positive cells contain frequent regions of single-stranded DNA. Furthermore, our data shows that the ratio of C-rich to G-rich reads in S1-END-seq data can distinguish ALT-negative (0.98–0.99) from ALT-positive cell lines (0.4–0.58). Based on this ratio, we estimate that the ALT-positive cells analyzed here contain a minimum of three single-stranded telomeric DNA regions per chromosome end (*Figure 4—figure supplement 1F and G*).

We also investigated whether reads detected by S1-END-seq are more likely to contain VTRs compared to the terminal portions of telomeres. We compared the frequency of VTRs within the first 30 nucleotides of the S1-END-seq reads from U2OS libraries (3.8%) to their control END-seq U2OS libraries (1.93%). This shows that the frequency of VTRs is higher in the internal portions of chromosome ends compared to the terminal portions (*Figure 4—figure supplement 1H*).

To determine whether the presence of ssDNA at telomeres in ALT cells is a feature intrinsic to these cells or a feature of the ALT-mediated telomere elongation, we measured the levels of ssDNA at telomeres in cells lacking BLM. Depletion of BLM is sufficient to suppress ALT-mediated telomere elongation, as shown by the loss of ALT-associated features and progressive telomere shortening (*Jiang et al., 2024*; *Loe et al., 2020*; *O'Sullivan et al., 2014*; *Min et al., 2019*; *Sobinoff et al., 2017*). As expected, we found similar levels of C-rich and G-rich reads in S1-END-seq data from U2OS cells (C/G strand ratio: 0.55, *Figure 4D*), indicating a minimum of 5 ssDNA regions per chromosome end. By contrast, BLM-deficient U2OS cells show a bias toward C-rich reads (C/G strand ratio: 0.7, *Figure 4D*), indicating that these cells have reduced levels of ssDNA at chromosome ends compared to the parental ALT-positive U2OS cell line (at least one ssDNA region per chromosome end). Given that BLM helicase contributes to telomere stability by resolving secondary structures and ensures proper telomeres replication regardless of telomere maintenance mechanism (*Barefield and Karlseder, 2012*; *Drosopoulos et al., 2015*; *Sfeir et al., 2009*), it is possible that ssDNA detected in the absence of BLM is due to replication fidelity. Contrary to this, most of the reads detected in both BLM-proficient and deficient cells by END-seq were coming from the C-strand (*Figure 4C*). As a control, we tested whether artificially introducing DSBs at telomeres using the TRF1-FokI construct (*Cho et al., 2014*) would abrogate the difference in BLM-proficient and -deficient cells. Expression of TRF1-Fok1 results in DSBs at telomeres, which are detected as the appearance of G-rich reads by END-seq in both BLM-proficient and BLM-deficient cells (*Figure 4G*). Expression of the TRF1-FokI nuclease in BLM-deficient cells was sufficient to induce single-stranded telomeric DNA at telomeres as seen by ssTelo signal (*Figure 4E and F*) as well as a C/G strand ratio close to what was seen in U2OS cells (*Figure 4H*).

Collectively, these data show that cells undergoing ALT have multiple single-stranded DNA regions that vastly exceed the number of chromosome ends. Accumulation of single-stranded telomeric DNA is tightly linked to the ALT process itself.

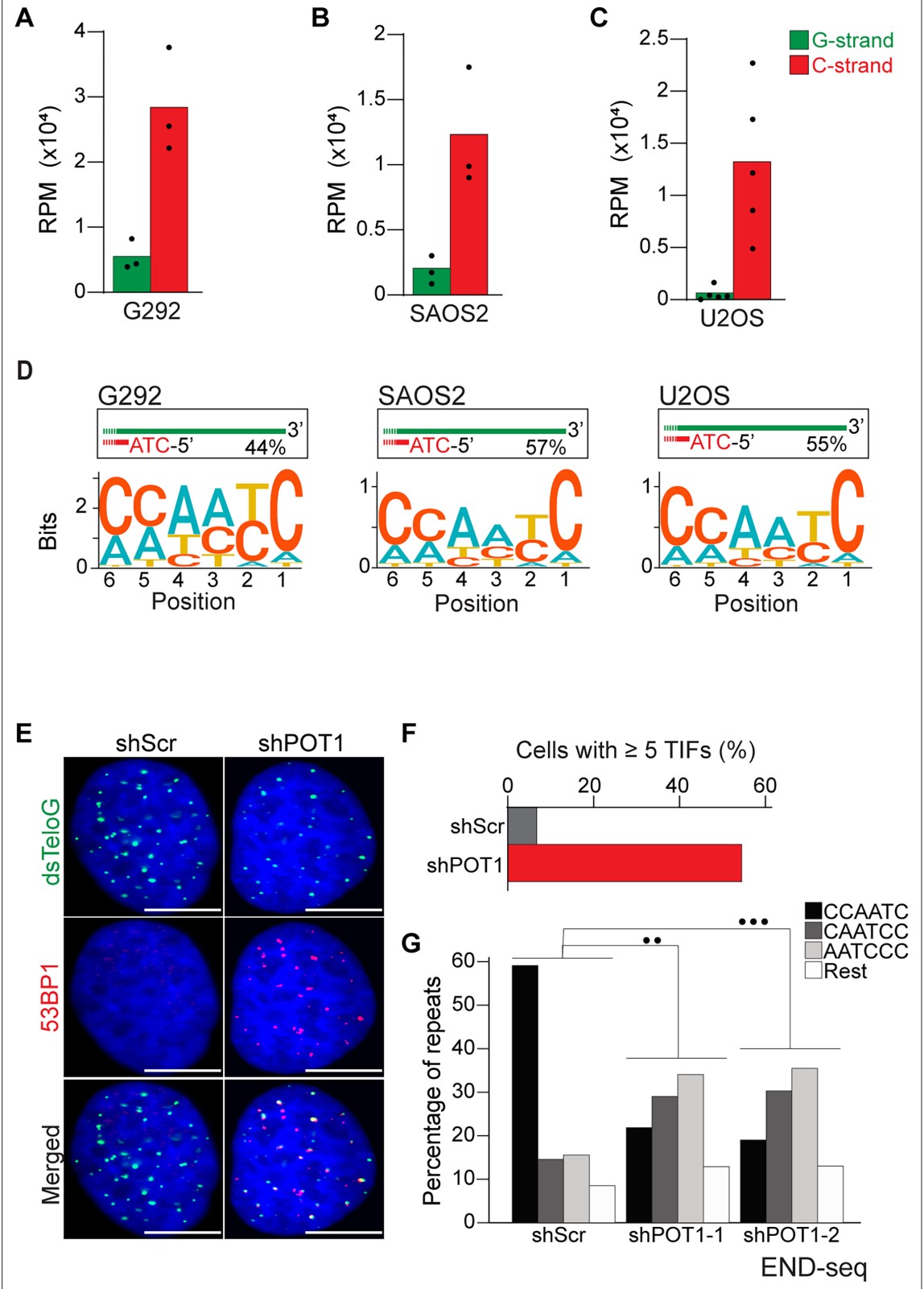

**Figure 3.** Alternative lengthening of telomeres (ALT) cells have precise 5′ termini. (**A–C**) Number of telomeric reads containing at least four consecutive telomeric repeats corresponding to the G-rich (green) and C-rich (red) strands in the ALT-positive G292, SAOS2, and U2OS cell lines. Telomeric reads are normalized to the total number of reads identified by END-seq and are represented as reads per million (RPM). (**D**) Sequence logo representing the conservation at chromosome ends. A sequence logo representing the conservation (bits) of the last 6 nucleotides at the 5′ end of chromosomes

*Figure 3 continued*

in ALT cells. Fraction of reads with CCAATC as 5′ end is displayed. (**E**) Cells expressing either a nontargeting shRNA (shScr) or a shRNA targeting POT1 (shPOT1-1) were stained for 53BP1 (red) and telomeric DNA (TTAGGG, green). The scale bar represents 10 μm. (**F**) Quantification of the data shown in (**E**). Graphs indicated the percentage of cells that have at least five telomere dysfunctional foci (TIF) with 53BP1 co-localizing at telomeres. (**G**) Percentage of telomeric reads that have the indicated sequence as a 5′ sequence that have as a 5′ end. The following sequences (CCCAAT-5′, TCCC AA-5′, and ATCCCA-5′) are grouped and labeled as 'Rest'. Cells expressing either a nontargeting shRNA (shScr) or a shRNA targeting POT1 (shPOT1-1 and shPOT1-2) were harvested 3 days post induction and analyzed by END-seq. Kullback–Leibler divergence (KL divergence) analysis was used to compare the distributions of the individual conditions. KL divergence between 0.25 and 0.375 is represented by two dots (●●), KL divergence greater than 0.375 is represented by three dots (●●●).

The online version of this article includes the following figure supplement(s) for figure 3:

**Figure supplement 1.** Analysis of 5′ termini and of presence of variant telomeric repeats in U2OS.

## Discussion

### END-seq as an unbiased tool to study the terminal telomeric repeat

Here, we show that END-seq can efficiently capture 5′ chromosome termini, providing a high-resolution and unbiased approach to studying the terminal portion of chromosome ends. We found that by END-seq, chromosome ends are detected as one-ended DSBs corresponding to the 5′ C-rich telomeric strand. Our data show that, in agreement with previous work (*Sfeir et al., 2005*), most chromosomes have the sequence ATC as the 5′ end. END-seq has several advantages over alternative approaches: it is unbiased since it does not rely on capture oligos, it can be applied to telomeres of any length, and does not require any prior knowledge of the subtelomeric sequence. While END-seq is designed to capture chromosome ends in an unbiased manner, we cannot exclude the possibility that certain telomeric structures may hinder detection, potentially leading to underrepresentation of a subset of telomeres. Nevertheless, we have been able to define the terminal identity of chromosome ends in cells with long telomeres, such as murine cells and ALT-positive cells, as well as cells with poorly characterized subtelomeric regions, such as canine cells. These data show that the ATC-5′ sequence appears to be universal, highlighting a conserved mechanism of end-processing in mammalian cells. We also examined the effect of the depletion of POT1 paralogs on the 5′ chromosome termini in mouse cells. While POT1a depletion altered the sequence at 5′ ends, POT1b depletion had no effect. This finding adds to the growing evidence that POT1a and POT1b have distinct functions at mouse telomeres (*Tesmer et al., 2023*; *Denchi and de Lange, 2007*; *Wu et al., 2006*; *Nurk et al., 2022*; *Hockemeyer et al., 2006*; *Gu et al., 2021*).

### Canonical POT1-dependent regulation of the terminal sequences in ALT-positive cells

Our data show that ALT-positive cells, which use recombination-based mechanisms for telomere elongation, have the same regulation of terminal repeats as telomerase-positive cells. END-seq analysis of three different ALT-positive cell lines (G292, SAOS2, and U2OS) revealed that the majority of telomeric reads correspond to C-rich reads starting with the canonical CCAATC-5′ sequence. This indicates that the regulation of the terminal 5′ end of chromosome ends occurs independently of the telomere elongation mechanism and that ALT-based recombination does not alter this process. Furthermore, our data show that POT1 is critical, even in U2OS cells, to define the 5′ end termini of telomeres.

### Single-stranded DNA in ALT telomeres

Using the unbiased S1-END-seq approach, we show that the presence of single-stranded telomeric DNA characterizes ALT-positive cells. This property of ALT is unique in that cells can be readily classified as ALT-positive or ALT-negative based on the occurrence of S1-cuts within the telomeric reads (*Figure 4B*). The observed single-stranded telomeric DNA may arise from various sources, including R-loops, D-loops, internal DNA loops (i-loops), and extrachromosomal circles, which are known to be prevalent in ALT (*Dilley et al., 2016*; *Henson et al., 2009*; *Mazzucco et al., 2020*; *Arora et al., 2014*).

Notably, our findings align with previous indirect evidence suggesting that ALT-positive cells exhibit increased levels of telomeric ssDNA, RPA-staining, and FISH under native conditions (*Loe et al., 2020*; *Grudic et al., 2007*; *Claude et al., 2021*; *Frank et al., 2022*). Based on our data, we can now conclude that these assays are all indeed detecting the telomeric ssDNA. Due to the nature of S1

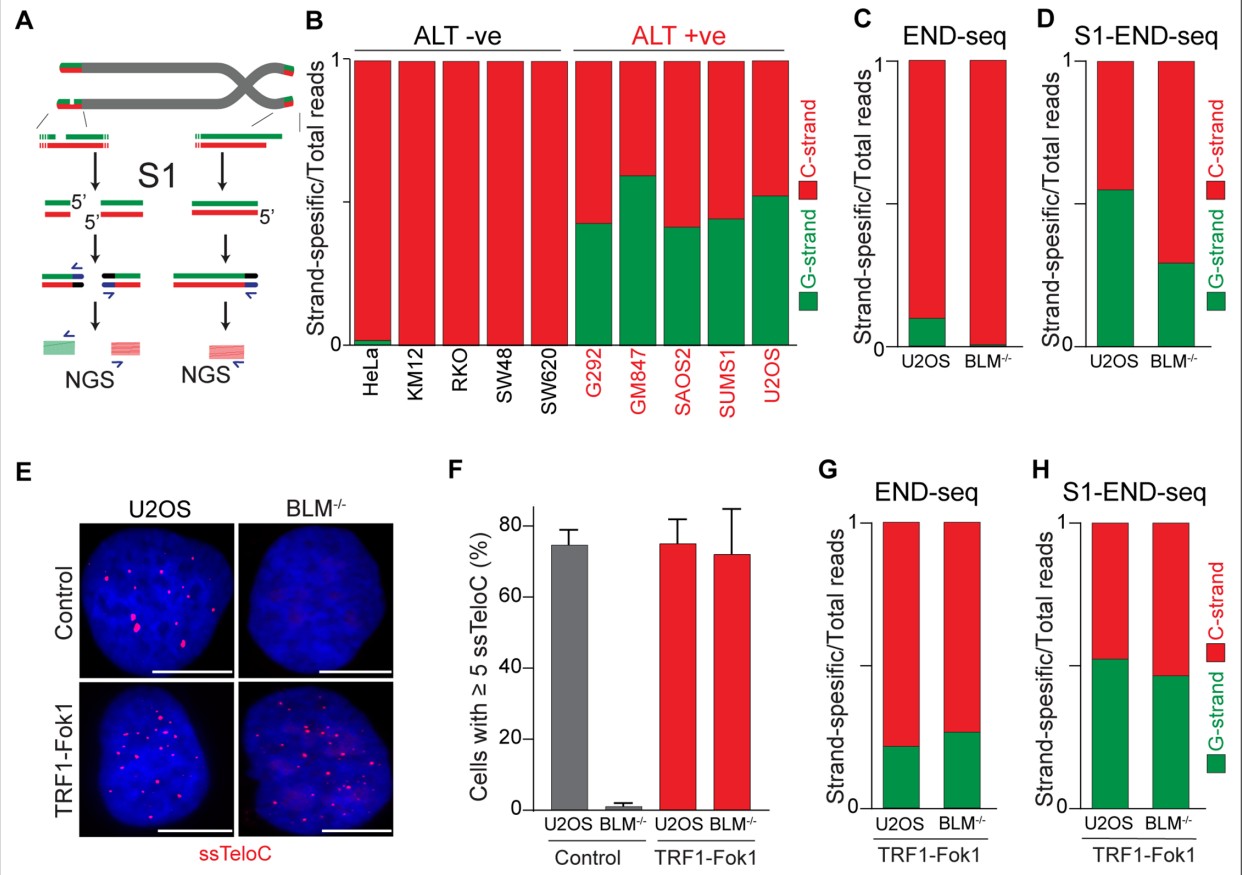

**Figure 4.** Alternative lengthening of telomeres (ALT) telomeres are readily distinguished due to presence of telomeric ssDNA by sequencing. (A) Schematic representation of the S1-END-seq method applied to telomeres containing either an internal ssDNA region (left) or only the natural G-rich overhang (right). S1 nuclease treatment cleaves ssDNA regions generating two-ended double-strand break (DSB) (left) or one-ended DSB (right). DNA ends are ligated to biotinylated adaptors, purified, and subsequently sequenced by NGS. Reads originating from the double-ended DSB align to either side of the break (left), resulting in C-rich (red) and G-rich (green) reads. Reads originating from the single-ended chromosome ends align only to the C-rich template (red). (B) Proportion of telomeric reads (containing at least four consecutive telomeric reads) corresponding to G-rich reads (green) or C-rich reads (red). (C, D) Proportion of telomeric reads (containing at least four consecutive telomeric reads) corresponding to G-rich reads (green) or C-rich reads (red) in BLM-proficient (U2OS) or -deficient (BML⁻/⁻) cells by END-seq (C) or S1-seq (D). (E) Native telomeric FISH (ssTelo) in BLM-proficient (U2OS) or -deficient (BML⁻/⁻) cells either left untreated (control) or expressing TRF1-Fok1. The scale bar represents 10 µm. (F) Quantification of the data shown in (E), cells with five ssTelo signal (or greater) were scored as positive. When indicated, cells were induced to express the TRF1-Fok1 nuclease for 24 hr prior to harvesting. (G, H) Proportion of telomeric reads (containing at least four consecutive telomeric reads) corresponding to G-rich reads (green) or C-rich reads (red) in BLM-proficient (U2OS) or -deficient (BML⁻/⁻) cells expressing TRF1-Fok1 by END-seq (G) or S1-seq (H).

The online version of this article includes the following figure supplement(s) for figure 4:

**Figure supplement 1.** ssDNA in alternative lengthening of telomeres (ALT) telomeres.

cleavage and the repetitive nature of telomeric reads, it is impossible to establish how many regions within telomeres have ssDNA or the average length of the single-stranded DNA tract within telomeres. However, based on the relative ratio between G-rich and C-rich strands, we estimate that there are at least five distinct regions of single-stranded DNA for each chromosome end in ALT-positive cells (*Figure 4—figure supplement 1G*). The origin of the observed single-stranded telomeric DNA remains to be determined. It is possible that, in ALT cells, partially single-stranded extrachromosomal DNA in the form of C-circles is captured by the S1-END-seq approach. Alternatively, we may be detecting intermediates of the ALT process at chromosome ends or single-stranded DNA generated by strand displacement due to the presence of R-loops and/or G-quadruplexes. Future studies will be needed to elucidate the mechanisms underlying the generation of ssDNA at telomeres in ALT cells. Nevertheless, our findings provide a rationale for using ssTelo as a biomarker for ALT-positive

cell detection, offering new avenues for studying and targeting ALT-driven cancers (*Loe et al., 2020*; *Claude et al., 2021*; *Frank et al., 2022*; *Quintanilla et al., 2025*).

# Materials and methods

## Key resources table

| Reagent type (species) or resource | Designation | Source or reference | Identifiers | Additional information |
|---|---|---|---|---|
| Cell line (*Homo sapiens*) | U2OS | ATCC | HTB-96, RRID:CVCL_0042 | |
| Cell line (*H. sapiens*) | HeLa 1.2.11 | *Takai et al., 2010* | RRID:CVCL_7908 | |
| Cell line (*H. sapiens*) | GM847 | *Henson et al., 2009* | RRID:CVCL_7908 | |
| Cell line (*H. sapiens*) | SUSM1 | *Henson et al., 2009* | RRID:CVCL_4903 | |
| Cell line (*H. sapiens*) | SAOS-2 | ATCC | HTB-85, RRID:CVCL_0548 | |
| Cell line (*H. sapiens*) | G292 | ATCC | CRL-1423, RRID:CVCL_2909 | |
| Cell line (*H. sapiens*) | U2OS BLM$^{-/-}$ | *Loe et al., 2020* | | |
| Cell line (*H. sapiens*) | HEK293T | ATCC | CRL-3216, RRID:CVCL_0063 | |
| Cell line (*Mus musculus*) | MEFs | *Okamoto et al., 2013* | | |
| Cell line (*M. musculus*) | mESC, E14 | This study | | See 'Materials and methods' |
| Cell line (*M. musculus*) | mESC, E14 *Pot1a::3×FLAG–FKBP12$^{F36V}$* | This study | | See 'Materials and methods' |
| Cell line (*M. musculus*) | mESC E14 *Pot1b$^{-/-}$* | This study | | See 'Materials and methods' |
| Cell line (*M. musculus*) | mESC E14 *Pot1a::3×FLAG–FKBP12$^{F36V}$ Pot1b$^{-/-}$* | This study | | See 'Materials and methods' |
| Cell line (*Canis lupus familiaris*) | KOS | This study | | See 'Materials and methods' |
| Antibody | Anti-TRF2 (rabbit polyclonal) | Novus Biologicals | Cat# NB110-57130, RRID:AB_844199 | IF (1:200) |
| Antibody | Anti-RPA32/RPA2 (4E4) (rat monoclonal) | Cell Signaling | Cat# 2208, RRID:AB_2238543 | IF (1:2000) |
| Antibody | Anti-Myc-Tag(9B11) (mouse monoclonal) | Cell Signaling | Cat# 2276, RRID:AB_331783 | IF (1:1000) |
| Antibody | Anti-53BP1 (rabbit polyclonal) | Novus Biologicals | Cat# NB100-304, RRID:AB_10003037 | IF (1:1000) |
| Recombinant DNA reagent | pRSIT16-U6Tet-sh-EF1-TetRep-2A-Puro | Cellecta | Cat# SVSHU6TEP-L | Plasmid |
| Recombinant DNA reagent | pLenti- CMV TRE3G Puro FLAG-DD-ER-mCherry-TRF1-FokI WT | *Dilley et al., 2016* | N/A | Plasmid |
| Sequence-based reagent | shPOT1 (TGTAGCTTGATCAGACACTTA) | IDT | N/A | shRNA |
| Sequence-based reagent | shPOT1-2 (TATGTATGCTAAATTGGATGG) | IDT | N/A | shRNA |
| Sequence-based reagent | hPOT1_F (qPCR primer) | IDT | N/A | 5'-CAGAACCTGACGACAGCTTTCC |
| Sequence-based reagent | hPOT1_R (qPCR primer) | IDT | N/A | 5'-GCACATAGTGGTGTCCTCTCCA |
| Sequence-based reagent | hGAPDH_F (qPCR primer) | IDT | N/A | 5'- GTCTCCTCTGACTTCAACAGCG |
| Sequence-based reagent | hGAPDH | IDT | N/A | 5'-ACCACCCTGTTGCTGTAGCCAA |
| Sequence-based reagent | mPOT1b_F (qPCR primer) | IDT | N/A | 5'-TTCGGCCCCAGTAGCACCTT |
| Sequence-based reagent | mPOT1b | IDT | N/A | 5'-TCTCTTGCTTAAAGTACGCAG |
| Sequence-based reagent | mGAPDH_F (qPCR primer) | IDT | N/A | 5'-TGTGTCCGTCGTGGATCTGA |
| Sequence-based reagent | mGAPDH | IDT | N/A | 5'-TTGCTGTTGAAGTCGCAGGAG |
| Sequence-based reagent | END-seq adaptor 1 | *Matos-Rodrigues et al., 2022* | | 5'-phosphate -GATCGGA AGAGCGTC G TGTAGGGAAAGAGTGUU[Biotin-dT]U[BiotindT]UUACACTC TTTCCCTACAC GACGCTCTTCCGATC*T-3' |
| Sequence-based reagent | END-seq adaptor 2 | *Matos-Rodrigues et al., 2022* | | 5'-phosphate -GATCGGA AGAGCACAC GTCUUUUUUUUAGACGTGTGCTCTTC CGA TC*T-3 |
| Sequence-based reagent | TelC-Cy3 | PNABio | Cat# F1002 | |

*Continued on next page*

*Continued*

| Reagent type (species) or resource | Designation | Source or reference | Identifiers | Additional information |
|---|---|---|---|---|
| Sequence-based reagent | TelG-Cy3 | PNABio | Cat# F1006 | |
| Commercial assay or kit | KAPA HiFi HotStart ReadyMix | Roche | Cat# KK2602 | |
| Commercial assay or kit | KAPA Library Quantification Kit | Roche | Cat# KK4824 | |
| Commercial assay or kit | RNeasy Plus Mini Kit | QIAGEN | Cat# 74104 | |
| Commercial assay or kit | SuperScript II Reverse Transcriptase | Thermo Fisher | Cat# 18064014 | |
| Commercial assay or kit | Power SYBR Green PCR Master | Fisher Scientific | Cat# 4368577 | |
| Commercial assay or kit | Qubit dsDNA HS assay kit | Thermo Fisher | Cat# Q32851 | |
| Other | Exonuclease T (ExoT) | NEB | Cat# M0265L | |
| Other | Klenow Fragment (3′→ 5′ exo-) | NEB | Cat# M0212L | |
| Other | Quick Ligation Kit | NEB | Cat# M2200L | |
| Other | USER enzyme | NEB | Cat# M5505L | |
| Other | T4 DNA Polymerase | NEB | Cat# M0203L | |
| Other | T4 Polynucleotide Kinase | NEB | Cat# M0201L | |
| Other | DNA Polymerase I, Large (Klenow) Fragment | NEB | Cat# M0210L | |
| Other | S1 Nuclease | Sigma-Aldrich | Cat# EN0321 | |
| Other | S1-END-seq KM12 | *Matos-Rodrigues et al., 2022* | | GEO # GSE203632 |
| Other | S1-END-seq RKO | *Matos-Rodrigues et al., 2022* | | GEO # GSE203632 |
| Other | S1-END-seq SW48 | *Matos-Rodrigues et al., 2022* | | GEO # GSE203632 |
| Other | S1-END-seq SW620 | *Matos-Rodrigues et al., 2022* | | GEO # GSE203632 |

## Methods

### Cell culture

Human, mouse, and dog cell lines were grown in DMEM medium supplemented with 10% fetal bovine serum (FBS) and 0.5 mg/ml penicillin, streptomycin, and L-glutamine (Gibco) and cultured at 37°C and 5% $CO_2$. HeLa cells used in this study were HeLa1.2.11. G292 cells were grown in McCoy medium supplement with 15% FBS and 0.5 mg/ml penicillin, streptomycin, and L-glutamine (Gibco). Cell lines were tested for mycoplasma contamination. BLM knock-out cells were previously described in *Loe et al., 2020*, mESC E14 and canine cell line KOS were kind gifts of the Scripps Research Institute Genomics Core Facility and Dr. Amy LeBlanc (NCI), respectively.

### Generation of mESC lines

Knockout and knock-in cell lines were generated via CRISPR/Cas9 gene targeting by transient transfection using Lipofectamine 2000 (Thermo Fisher Scientific). Cells were transfected with a plasmid encoding SpCas9 (px330) and a plasmid encoding specific gRNAs (backbone Addgene 41824). To introduce the dTAG13 controllable FKBP-tag POT1a in mESCs via homologous recombination-mediated repair, a synthetic repair template containing homology arms to the *Pot1a* locus, a blasticidin resistance marker, a 3xFLAG tag, and an FKBP tag was used, along with 2 sgRNA targeting the 5′ region of *Pot1a* (guide 1: 5′- TTACATTAGTGCTAATCCAA-3′, and guide 2: 5′-TTTTCATCAAACAATG TCTT-3′). To generate POT1b[-/-] cells, two gRNAs were used: guide 1 (5′-GTTCCTTTGCTTAAGTACGG -3′) and guide 2 (5′-GACATTTGATGGCACCTTAG-3′). Transfected cells were single-cell cloned, and resulting clones were screened by PCR to confirm the desired genomic deletion, which was further validated by Sanger sequencing.

## Lentiviral-mediated gene manipulation

Lentiviral particles were generated by transfection into HEK293T followed by isolation and transduction as previously described (*Loe et al., 2020*). Lentiviral expression was achieved with the following constructs and procedure:

### POT1 depletion

sgRNA targeting POT1 (POT1-1: TGTAGCTTGATCAGACACTTA, POT1-2: TATGTATGCTAAATTG GATGG) were cloned into an inducible lentiviral vector (pRSITEP-U6Tet-sh-EF1-TetRep-2A-Puro) (Cellecta). Following infection, cells were treated with 1 µg/ml of doxycycline to induce shRNA expression and harvested 3 days later.

### TRF1-Fok1 expression

pLenti-CMV-TRE3G-ER-TRF1-Fok1 expressing plasmids (*Dilley et al., 2016*) were a kind gift from Agnel Sfeir. Infected cells were treated with 1 µg/ml of doxycycline and 0.6 µM of 4-hydroxytamoxifen (4-OHT) to induce TRF1-Fok1 and harvested 24 hr later.

## RNA isolation and qRT-PCR

Total RNA was extracted from cells using the RNeasy Plus Mini Kit (QIAGEN), and cDNA was made using SuperScript II Reverse Transcriptase (Thermo Fisher). qPCR was performed using Power SYBR Green PCR Master (Fisher Scientific). Samples were run on a Bio-Rad CFX Opus 96 and analyzed on Bio-Rad CFX Maestro. The following primers were used in this study:

> hPOT1: 5′-CAGAACCTGACGACAGCTTTCC and 5′-GCACATAGTGGTGTCCTCTCCA.
> hGAPDH: 5′-GTCTCCTCTGACTTCAACAGCG and ACCACCCTGTTGCTGTAGCCAA.
> mPOT1b: 5′-TTCGGCCCCAGTAGCACCTT and 5′-TCTCTTGCTTAAAGTACGCAG.
> mGAPDH: 5′-TGTGTCCGTCGTGGATCTGA and 5′-TTGCTGTTGAAGTCGCAGGAG.

## Immunofluorescence (IF) and FISH-based imaging

IF, IF-FISH, and ssTelo staining were carried out as described previously (*Loe et al., 2020*; *Azeroglu et al., 2024*).

PNA probes used in this study were TelC-Cy3 (PNABio, Cat# F1002) and TelG-Cy3 PNA probe (PNABio, Cat# F1006).

Primary antibodies used in this study were TRF2 (NB110-57130, Novus Biologicals, rabbit), RPA32/ RPA2 (2208, Cell Signaling, rat), Myc (2276, Cell Signaling, mouse), and 53BP1 (NB100-304, Novus Biologicals, rabbit).

## Imaging and image analysis

Images were acquired using a Zeiss Axio Imager M2 with an Axiocam 702 camera and ZEN 2.6 (blue edition) software; for each experiment and condition, a minimum of 250 cells were imaged. Final figures were assembled using Adobe Photoshop 25.7.0 and Adobe Illustrator 28.5.

## END-seq and S1-END-seq

For END-seq and S1-END-seq, $6–20 \times 10^6$ cells were collected, embedded in 1% agarose plugs, and processed as previously reported (*Canela et al., 2016*; *Matos-Rodrigues et al., 2022*). For mouse spleen library, mice were euthanized with $CO_2$ following ethical guidelines, and the spleens were surgically removed. Spleen tissues were mashed through a 70 µm cell strainer to obtain a single-cell suspension. The suspension was then centrifuged for 5 min at $300 \times g$, the pellet was washed with ice-cold PBS three times before proceeding with library preparation. Briefly, plugs were incubated with proteinase K solution for 1 hr at 50°C and then for 7 hr at 37°C, followed by consecutive washes in a wash buffer (10 mM Tris, pH 8.0, and 50 mM EDTA) and TE (10 mM Tris, pH 8.0, and 1 mM EDTA). Washed plugs were then treated with RNAseA (Puregene, QIAGEN), washed again in Wash Buffer, and stored at 4°C for up to 2 weeks. End blunting, A-tailing, and ligation to biotinylated hairpin adaptor 1 (ENDseq-adaptor-1, 5′-Phos-GATCGGAAGAGCGTCGTGTAGGGAAAGAGTGU U[Biotin-dT]U[Biotin-dT]UUACACTCTTTCCCTACACGACGCTCTTCCGATC∗T-3′ [∗phosphorothioate

bond]) were performed in the plug to minimize shearing and in vitro damage. Following this, high molecular weight DNA was isolated from agarose plugs and fragmented by sonication. Subsequently, adaptor-ligated fragments were captured with streptavidin-coated beads and sonication-created ends were repaired, A-tailed, and ligated with a distal adaptor (ENDseq-adaptor-2, 5'-Phos-GATCGGAA GAGCACACGTCUUUUUUUUAGACGTGTGCTCTTCCGATC∗T-3' [∗phosphorothioate bond]). Subsequent PCR amplification resulted in ready-to-sequence libraries where the sequenced first base corresponds precisely to the first base of the blunted DSB on either side of the break. For S1-END-seq, proteinase K and RNAseA-treated plugs were treated with 100 U of S1 nuclease (Thermo Fisher) at 37°C for 30 min and the reaction was terminated by the addition of EDTA to a final concentration of 10 mM. S1-treated plugs were then processed through the standard END-seq protocol. For T7 treatment, proteinase K and RNAseA-treated plugs were incubated with 50 U of T7 exonuclease (NEB) at 37°C for 30 min before processing through the standard END-seq protocol.

The libraries were sequenced on the Nextseq 550 or Nextseq 2000 platform (Illumina) using 75-bp single-end read kits. For each treatment condition, the control (nontreated) and sample libraries were sequenced in parallel in the same flow cell.

## Telomeric motif detector and quantifier

We developed a Python script, 'getTelomereENDseq', to analyze gzipped FASTQ files for the presence of telomeric motifs in DNA sequences on a command-line environment using a Linux-based operating system and the statistical environment RStudio. The script identifies sequences containing G-rich and C-rich telomeric repeats, quantifies their occurrences, and calculates their relative frequencies. It takes a gzipped FASTQ file as input, and the minimum number of telomeric repeats is required. It generates two output files: one with sequences containing G-rich motifs and another with sequences containing C-rich motifs. For this study, we used a minimum of four consecutive TTAGGG repeats for G-rich reads and a minimum of four consecutive CCCTAA repeats for C-rich reads. Following this, in RStudio, using the *stringi*, *ggseqlogo*, *ggplot2*, *ggsci*, and *tidyverse* packages, the telomeric reads were trimmed by the last six nucleotides, reversed, and organized into data frames. These data frames were then used to plot the frequencies of telomeric repeats or motifs.

## Statistical analysis and data visualization

We used RStudio and GraphPad Prism v10 for data visualization and statistical analysis. Statistical analyses are specified in figure legends. For microscopy data, $p < 0.05$ was considered statistically significant, and p values were assessed by unpaired parametric *t*-test. The sample size was not predetermined. Statistical analyses are specified in the figure legends. We used KL divergence to compare the distributions of frequencies of telomeric repeats between different samples. The KL divergence is a measure of the difference between two probability distributions. It quantifies how one probability distribution diverges from a second, expected distribution. To compute KL divergence, we used the philentropy package in RStudio. For this study, KL divergence values less than 0.125 are considered nonsignificant, and the rest assigned as follows: ● indicates KL divergence value between 0.125 and 0.25, ●● indicates a value between 0.25 and 0.375, and ●●● indicates a value greater than 0.375.

# Acknowledgements

We are grateful to Agnel Sfeir, Amy LeBlanc, and Gianna Tricola for providing reagents. We thank the CCR Genomic Core for assisting with the NGS experiments performed in this study. We thank members of the Lazzerini Denchi laboratory, Sam John, Niek van Wietmarschen, and Travis Stracker for critical feedback on this work. This work utilized the computational resources of the NIH HPC Biowulf cluster (https://hpc.nih.gov). Research in the ELD lab is funded by the NIH Intramural Research Program of the National Institutes of Health (NIH), National Cancer Institute (NCI), and Center for Cancer Research, project 1-ZIA-BC011815-03.

# Additional information

## Funding

| Funder | Grant reference number | Author |
|--------|------------------------|--------|
| National Institutes of Health | 1-ZIA-BC011815-03 | Eros Lazzerini-Denchi |

The funders had no role in study design, data collection and interpretation, or the decision to submit the work for publication.

## Author contributions

Benura Azeroglu, Conceptualization, Data curation, Formal analysis, Validation, Investigation, Visualization, Methodology, Writing – original draft, Writing – review and editing; Wei Wu, Raphael Pavani, Methodology; Ranjodh Singh Sandhu, Tadahiko Matsumoto, Resources; André Nussenzweig, Resources, Methodology; Eros Lazzerini-Denchi, Conceptualization, Resources, Supervision, Funding acquisition, Investigation, Writing – original draft, Writing – review and editing

## Author ORCIDs

Benura Azeroglu  https://orcid.org/0000-0001-7243-384X
Raphael Pavani  https://orcid.org/0000-0002-7187-2995
Ranjodh Singh Sandhu  https://orcid.org/0009-0008-2340-3423
Eros Lazzerini-Denchi  https://orcid.org/0000-0001-5378-4644

Reviewer #1 (Public review): https://doi.org/10.7554/eLife.106657.3.sa1
Reviewer #2 (Public review): https://doi.org/10.7554/eLife.106657.3.sa2
Reviewer #3 (Public review): https://doi.org/10.7554/eLife.106657.3.sa3
Author response https://doi.org/10.7554/eLife.106657.3.sa4

# Additional files

## Supplementary files
MDAR checklist

## Data availability

The accession number for the datasets reported in this paper is available at GEO with accession number: GSE301770.

The following dataset was generated:

| Author(s) | Year | Dataset title | Dataset URL | Database and Identifier |
|-----------|------|---------------|-------------|-------------------------|
| Azeroglu B, Wu W, Pavani R, Sandhu R, Matsumoto T, Nussenzweig A, Lazzerini Denchi E | 2025 | Conserved and Unique Features of Terminal Telomeric Sequences in ALT-Positive Cancer Cells | https://www.ncbi.nlm.nih.gov/geo/query/acc.cgi?acc=GSE301770 | NCBI Gene Expression Omnibus, GSE301770 |

The following previously published dataset was used:

| Author(s) | Year | Dataset title | Dataset URL | Database and Identifier |
|-----------|------|---------------|-------------|-------------------------|
| Matos-Rodrigues G, van Wietmarschen N, Wu W, Tripathi V, Koussa N, Pavani R, Callen E, Nathan W, Belinky F, Mohammed A, Napierala M, Usdin K, Ansari AZ, Mirkin SM, Nussenzweig A | 2022 | Triplexes are Dynamic DNA Secondary Structures Formed in vivo and Linked to Genome Instability [END-seq] | https://www.ncbi.nlm.nih.gov/geo/query/acc.cgi?acc=GSE203632 | NCBI Gene Expression Omnibus, GSE203632 |

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
