## [Editor Report · eLife Assessment]

This study demonstrates the application of END-seq, originally developed to study genomewide DNA double-strand breaks, to telomere biology; the work packs a punch, concisely demonstrating the utility of this approach and the new insights that can be gained. The authors confirm that telomeres in telomerase-positive cells terminate with 5'-ATC in a Pot1-dependent manner and demonstrate that this principle holds true in telomerase-negative ALT cells as well. S1-END-seq is similarly developed for telomeres, showing that ALT cells harbor several regions of ssDNA. The study is well-executed and **convincing**, the new insights are **fundamental** and **compelling**, and the optimized END-seq approaches will be widely utilized. The work will prompt additional studies that the reviewers look forward to, including combining telomeric END-seq with long-read sequencing to address the distribution and origin of variant telomere repeats and ssDNA along telomeres in ALT and telomerase-positive settings.

---

## [Referee Report · Reviewer #1 (Public review)]

Summary

This manuscript from Azeroglu et al. presents the application of END-Seq to examine the sequence composition of chromosome termini, i.e., telomeres. END-seq is a powerful genome sequencing strategy developed in Andre Nussesweig's lab to examine the sequences at DNA break sites. Here, END-Seq is applied to explore the nucleotide sequences at telomeres and to ascertain (i) whether the terminal end sequence is conserved in cells that activate ALT telomere elongation mechanism and (ii) whether the processes responsible for telomere end sequence regulation are conserved. With these aims clearly articulated, the authors convincingly show the power of this technique to examine telomere end-processing.

Strengths

(1) The authors effectively demonstrate the application of END-seq for these purposes. They verify prior data that 5'terminal sequences of telomeres in Hela and RPE cells end in a canonical ATC sequence motif. They verify that the same sequence is present at the 5' ends of telomeres by performing END-seq across a panel of ALT cancer cells. As in non-ALT cells, the established role of POT1, a ssDNA telomere binding protein, in coordinating the mechanism that maintains the canonical ATC motif is likewise verified. However, by performing END-Seq in mouse cells lacking POT1 isoforms, POT1a and POT1b, the authors uncover that POT1b is dispensable for this process. This reveals a novel, important insight relating to the evolution of POT1 as a telomere regulatory factor.

(2) The authors then demonstrate the utility of S1-END-seq, a variation of END-Seq, to explore the purported abundance of single-stranded DNA at telomeres within telomeres of ALT cancer cells. Here, they demonstrate that ssDNA abundance is an intrinsic aspect of ALT telomeres and is dependent on the activity of BLM, a crucial mediator of ALT.

Overall, the authors have effectively shown that END-seq can be applied to examine processes maintaining telomeres in normal and cancerous cells across multiple species. Using END-Seq, the authors confirm prior cell biological and sequencing data and the role of POT1 and BLM in regulating telomere termini sequences and ssDNA abundance. The study is nice and well-written, with the experimental rationale and outcomes clearly explained.

Weaknesses

This reviewer finds little to argue with in this study. It is timely and highly valuable for the telomere field. One minor question would be whether the authors could expand more on the application of END-Seq to examine the processive steps of the ALT mechanism? Can they speculate if the ssDNA detected in ALT cells might be an intermediate generated during BIR (i.e., is the ssDNA displaced strand during BIR) or a lesion? Furthermore, have the authors assessed whether ssDNA lesions are due to the loss of ATRX or DAXX, either of which can be mutated in the ALT setting?

Comments on revisions:

The authors addressed the comments. Thank you.

---

## [Referee Report · Reviewer #2 (Public review)]

This is a short yet very clear manuscript demonstrating that two methods (END-seq and S1-END-seq), previously developed in the Nussenzweig laboratory to study DSBs in the genome, can also be applied to the 5' ends of mammalian telomeres and the accumulation of telomeric single-stranded DNA.

The authors first validate the applicability of END-seq using different approaches and confirm that mammalian telomeres preferentially end with an ATC 5' end through a mechanism that requires intact POT1 (POT1a in mice). They then extend their analysis to cells that maintain telomeres through the ALT mechanism and demonstrate that, in these cells as well, telomeres frequently end in an ATC 5' sequence via a POT1-dependent mechanism. Using S1-END-seq, the authors further show that ALT telomeres contain single-stranded DNA and estimate that each telomere in ALT cells harbors at least five regions of ssDNA.

I find this work very interesting and incisive. It clearly demonstrates that END-seq can be applied with unprecedented depth and precision to the study of telomeric features such as the 5' end and ssDNA. The data are very clear and thoroughly interpreted, and the manuscript is well written. The results are carefully analyzed and effectively presented. Overall, I find this manuscript worthy of publication, as the optimized END-seq methods described here will likely be widely utilized in the telomere field.

Also, the authors have satisfactorily addressed my previous comments.

---

## [Referee Report · Reviewer #3 (Public review)]

Summary:

A subset of cancer cells attain replicative immortality by activating the ALT mechanism of telomere maintenance, which is currently the subject of intense research due to its potential for novel targeted therapies. Key questions remain in the field, such as whether ALT telomeres adhere to the same end-protection rules as telomeres in telomerase-expressing cells, or if ALT telomeres possess unique properties that could be targeted with new, less toxic cancer therapies. Both questions, along with the approaches developed by the authors to address them, are highly relevant.

Strengths:

Since chromosome ends resemble one-ended DSBs, the authors hypothesized that the previously described END-SEQ protocol could be used to accurately sequence the 5' end of telomeres on the C-rich strand. As expected, most reads corresponded to the C-rich strand and, confirming previous observation by the de Lange's group, most chromosomes end with the ATC-5' sequence, a feature that was found to be dependent on POT1 and to be conserved in both human ALT cells and mouse cells. Through a complementary method, S1-END-SEQ, the authors further explored ssDNA regions at telomeres, providing new insights into the characteristics of ALT telomeres. The study is original, the experiments were well-controlled and excellently executed.

Weaknesses:

A few additional experiments would have strengthened the results such as combining error-free long-read sequencing with END-SEQ to compare the abundance of VTRs within telomeres versus at their distal ends.

Along this line, are VTRs increased at ssDNA regions of ALT telomeres? What is the frequency of VTRs in the END-SEQ analysis of TRF1-FokI-expressing ALT cells? Is it also increased? Has TRF1-FokI been applied to telomerase-expressing cells to compare VTR frequencies at internal sites between ALT and telomerase-expressing cells?

To what extent do ECTRs contribute to telomeric ssDNA?

Future experiments may help shed light on this

---

## [Author Response]

The following is the authors’ response to the original reviews.

**Reviewer #1 (Recommendations for the authors):**
One minor question would be whether the authors could expand more on the application of END-Seq to examine the processive steps of the ALT mechanism? Can they speculate if the ssDNA detected in ALT cells might be an intermediate generated during BIR (i.e., is the ssDNA displaced strand during BIR) or a lesion? Furthermore, have the authors assessed whether ssDNA lesions are due to the loss of ATRX or DAXX, either of which can be mutated in the ALT setting?

We appreciate the reviewer’s insightful questions regarding the application of our assays to investigate the nature of the ssDNA detected in ALT telomeres. Our primary aim in this study was to establish the utility of END-seq and S1-END-seq in telomere biology and to demonstrate their applicability across both ALT-positive and -negative contexts. We agree that exploring the mechanistic origins of ssDNA would be highly informative, and we anticipate that END-seq–based approaches will be well suited for such future studies. However, it remains unclear whether the resolution of S1-END-seq is sufficient to capture transient intermediates such as those generated during BIR. We have now included a brief speculative statement in the revised discussion addressing the potential nature of ssDNA at telomeres in ALT cells.

**Reviewer #2 (Recommendations for the authors):**
How can we be sure that all telomeres are equally represented? The authors seem to assume that END-seq captures all chromosome ends equally, but can we be certain of this? While I do not see an obvious way to resolve this experimentally, I recommend discussing this potential bias more extensively in the manuscript.

We thank the reviewer for raising this important point. END-seq and S1-END-seq are unbiased methods designed to capture either double-stranded or single-stranded DNA that can be converted into blunt-ended double-stranded DNA and ligated to a capture oligo. As such, if a subset of telomeres cannot be processed using this approach, it is possible that these telomeres may be underrepresented or lost. However, to our knowledge, there are no proposed telomeric structures that would prevent capture using this method. For example, even if a subset of telomeres possesses a 5′ overhang, it would still be captured by END-seq. Indeed, we observed the consistent presence of the 5′-ATC motif across multiple cell lines and species (human, mouse, and dog). More importantly, we detected predictable and significant changes in sequence composition when telomere ends were experimentally altered, either in vivo (via POT1 depletion) or in vitro (via T7 exonuclease treatment). Together, these findings support the robustness of the method in capturing a representative and dynamic view of telomeres across different systems.

That said, we have now included a brief statement in the revised discussion acknowledging that we cannot fully exclude the possibility that a subset of telomeres may be missed due to unusual or uncharacterized structures

I believe Figures 1 and 2 should be merged.

We appreciate the reviewer’s suggestion to merge Figures 1 and 2. However, we feel that keeping them as separate figures better preserves the logical flow of the manuscript and allows the validation of END-seq and its application to be presented with appropriate clarity and focus. We hope the reviewer agrees that this layout enhances the clarity and interpretability of the data.

Scale bars should be added to all microscopy figures.

We thank the reviewer for pointing this out. We have now added scale bars to all the microscopy panels in the figures and included the scale details in the figure legends.

**Reviewer #3 (Recommendations for the authors):**
Overall, the discussion section is lacking depth and should be expanded and a few additional experiments should be performed to clarify the results.

We thank the reviewer for the suggestions. Based on this reviewer’s comments and comments for the other reviewers, we incorporated several points into the discussion. As a result, we hope that we provide additional depth to our conclusions.

(1) The finding that the abundance of variant telomeric repeats (VTRs) within the final 30 nucleotides of the telomeric 5' ends is similar in both telomerase-expressing and ALT cells is intriguing, but the authors do not address this result. Could the authors provide more insight into this observation and suggest potential explanations? As the frequency of VTRs does not seem to be upregulated in POT1-depleted cells, what then drives the appearance of VTRs on the C-strand at the very end of telomeres? Is CST-Pola complex responsible?

The reviewer raises a very interesting and relevant point. We are hesitant at this point to speculate on why we do not see a difference in variant repeats in ALT versus non-ALT cells, since additional data would be needed. One possibility is that variant repeats in ALT cells accumulate stochastically within telomeres but are selected against when they are present at the terminal portion of chromosome ends. However, to prove this hypothesis, we would need error-free long-read technology combined with END-seq. We feel that developing this approach would be beyond the scope of this manuscript.

(2) The authors also note that, in ALT cells, the frequency of VTRs in the first 30 nucleotides of the S1-END-SEQ reads is higher compared to END-SEQ, but this finding is not discussed either. Do the authors think that the presence of ssDNA regions is associated with the VTRs? Along this line, what is the frequency of VTRs in the END-SEQ analysis of TRF1-FokI-expressing ALT cells? Is it also increased? Has TRF1-FokI been applied to telomerase-expressing cells to compare VTR frequencies at internal sites between ALT and telomerase-expressing cells?

Similarly to what is discussed above, short reads have the advantage of being very accurate but do not provide sufficient length to establish the relative frequency of VTRs across the whole telomere sequence. The TRF1-FokI experiment is a good suggestion, but it would still be biased toward non-variant repeats due to the TRF1-binding properties. We plan to address these questions in a future study involving long-read sequencing and END-seq capture of telomeres.

Finally, in these experiments (S1-END-SEQ or END-SEQ in TRF1-Fok1), is the frequency of VTRs the same on both the C- and the G-rich strands? It is possible that the sequences are not fully complementary in regions where G4 structures form.

We thank the reviewer for this observation. While we do observe a higher frequency of variant telomeric repeats (VTRs) in the first 30 nucleotides of S1-END-seq reads compared to END-seq in ALT cells, we are currently unable to determine whether this difference is significant, as an appropriate control or matched normalization strategy for this comparison is lacking. Therefore, we refrain from overinterpreting the biological relevance of this observation.

The reviewer is absolutely correct. Our calculation did not exclude the possibility of extrachromosomal DNA as a source of telomeric ssDNA. We have now addressed this point in our discussion.

The reviewer is correct in pointing out that we still do not know what causes ssDNA at telomeres in ALT cells. Replication stress seems the most logical explanation based on the work of many labs in the field. However, our data did not reveal any significant difference in the levels of ssDNA at telomeres in non-ALT cells based on telomere length. We used the HeLa1.2.11 cell line (now clarified in the Materials section), which is the parental line of HeLa1.3 and has similarly long telomeres (~20 kb vs. ~23 kb). Despite their long telomeres and potential for replication-associated challenges such as G-quadruplex formation, HeLa1.2.11 cells did not exhibit the elevated levels of telomeric ssDNA that we observed in ALT cells (Figure 4B). Additional experiments are needed to map the occurrence of ssDNA at telomeres in relation to progression toward ALT.

(3) Based on the ratio of C-rich to G-rich reads in the S1-END-SEQ experiment, the authors estimate that ALT cells contain at least 3-5 ssDNA regions per chromosome end. While the calculation is understandable, this number could be discussed further to consider the possibility that the observed ratios (of roughly 0.5) might result from the presence of extrachromosomal DNA species, such as C-circles. The observed increase in the ratio of C-rich to G-rich reads in BLM-depleted cells supports this hypothesis, as BLM depletion suppresses C-circle formation in U2OS cells. To test this, the authors should examine the impact of POLD3 depletion on the C-rich/G-rich read ratio. Alternatively, they could separate high-molecular-weight (HMW) DNA from low-molecular-weight DNA in ALT cells and repeat the S1-END-SEQ in the HMW fraction.

The reviewer is absolutely correct. Our calculation did not exclude the possibility of extrachromosomal DNA as a source of telomeric ssDNA. We have now addressed this point in our discussion.

(4) What is the authors' perspective on the presence of ssDNA at ALT telomeres? Do they attribute this to replication stress? It would be helpful for the authors to repeat the S1-END-SEQ in telomerase-expressing cells with very long telomeres, such as HeLa1.3 cells, to determine if ssDNA is a specific feature of ALT cells or a result of replication stress. The increased abundance of G4 structures at telomeres in HeLa1.3 cells (as shown in J. Wong's lab) may indicate that replication stress is a factor. Similar to Wong's work, it would be valuable to compare the C-rich/G-rich read ratios in HeLa1.3 cells to those in ALT cells with similar telomeric DNA content.

The reviewer is correct in pointing out that we still do not know what causes ssDNA at telomeres in ALT cells. Replication stress seems the most logical explanation based on the work of many labs in the field. However, our data did not reveal any significant difference in the levels of ssDNA at telomeres in non-ALT cells based on telomere length. We used the HeLa1.2.11 cell line (now clarified in the Materials section), which is the parental line of HeLa1.3 and has similarly long telomeres (~20 kb vs. ~23 kb). Despite their long telomeres and potential for replication-associated challenges such as G-quadruplex formation, HeLa1.2.11 cells did not exhibit the elevated levels of telomeric ssDNA that we observed in ALT cells (Figure 4B). Additional experiments are needed to map the occurrence of ssDNA at telomeres in relation to progression toward ALT.

Finally, Reviewer #3 raises a list of minor points:

(1) The Y-axes of Figure 4 have been relabeled to account for the G-strand reads.

(2) Statistical analyses have been added to the figures where applicable.

(3) The manuscript has been carefully proofread to improve clarity and consistency throughout the text and figure legends

(4) We have revised the text to address issues related to the lack of cross-referencing between the supplementary figures and their corresponding legends.